# Fractional Order Model of the Two Dimensional Heat Transfer Process

**Krzysztof Oprzędkiewicz** *,† , **Wojciech Mitkowski** † and **Maciej Rosół** †

Department of Automatic Control and Robotics, AGH University, al. A. Mickiewicza 30, 30 023 Krakow, Poland;
wojciech.mitkowski@agh.edu.pl (W.M.); mr@agh.edu.pl (M.R.)
* Correspondence: kop@agh.edu.pl
† These authors contributed equally to this work.

**Abstract:** In this paper, a new, state space, fractional order model of a heat transfer in two dimensional plate is addressed. The proposed model derives directly from a two dimensional heat transfer equation. It employs the Caputo operator to express the fractional order differences along time. The spectrum decomposition and stability of the model are analysed. The formulae of impluse and step responses of the model are proved. Theoretical results are verified using experimental data from thermal camera. Comparison model vs experiment shows that the proposed fractional model is more accurate in the sense of MSE cost function than integer order model.

**Keywords:** fractional order systems; two dimensional heat transfer; fractional order state equation; Caputo operator; thermal camera





## 1. Introduction

It is known that the non integer order calculus can be applied in modeling of processes and phenomena hard to analyse with the use of other tools. Non integer order (NIO) or fractional order (FO) models of different physical phenomena have been presented by many authors. The amount of FO models of various processes is collected in the book [1]. The book [2] presents fractional order models of chaotic systems and Ionic Polymer Metal Composites (IPMC). Fractional models of ultracapacitors are "classics" of FO modeling. They are given for example by [3]. Distributed parameter systems can be also described using FO approach. As an example diffusion processes discussed in [4–6], can be given. A collection of recent results employing a new Atangana–Baleanu operator can be found in [7]. In this book, i.e., the FO blood alcohol model, the Christov diffusion equation and fractional advection-dispersion equation for groundwater transport process are presented.

Different thermal processes can also be described using FO approach. For example a temperature–heat flux relationship for heat flow in semi-infinite conductor is presented in [1], the beam heating problem is given in [3]. Theoretical analysis of fractional thermal equation is given, e.g., in [8–11].

Models of temperature fields obtained using thermal cameras are presented by [12,13], temperature models in three dimensional solid body are given in [14]. The use of fractional order approach to the modeling and control of heat systems is also presented in [15]. The analytical solution of the two dimensional heat equation is proposed in the paper [16]. Numerical methods of solution partial differential equation can be found, e.g., in [17]. Fractional Fourier integral operators are analyzed, u.a., in [18].

This paper is devoted to present a new, discrete, FO model of heat transfer process in two dimensional plate. Such a process is described by a partial differential equation (PDE) of parabolic type. The partial derivative along time is expressed by Caputo operator, both partial, spatial derivatives are integer order. Such a model of thermal process has not been previously published.

The paper is organized as follows: Preliminaries recall some elementary ideas and definitions from fractional calculus. Next, the state space model of two dimensional heat transfer process is proposed and discussed. Finally the experimental validation of theoretical results is given.

## 2. Preliminaries

At the beginning the non integer order, integro-differential operator is presented (see, e.g., [1,19–21]).

**Definition 1** (The elementary non integer order operator)**.** *The non integer order integro-differential operator is defined as follows:*

$$
{}_a D_t^\alpha g(t) = \begin{cases} \frac{d^\alpha g(t)}{dt^\alpha} & \alpha > 0 \\ g(t) & \alpha = 0 \\ \int_a^t g(\tau)(d\tau)^\alpha & \alpha < 0 \end{cases}.
\tag{1}
$$

*where a and t denote time limits for operator calculation, $\alpha \in \mathbb{R}$ denotes the non integer order of the operation.*

Next, an idea of complete Gamma Euler function is recalled (see for example [20]):

**Definition 2** (The complete Gamma function)**.**

$$
\Gamma(x) = \int_0^\infty t^{x-1} e^{-t} dt.
\tag{2}
$$

An idea of Mittag–Leffler function needs to be given next. It is a non integer order generalization of exponential function $e^{\lambda t}$ and it plays crucial role in the solution of fractional order (FO) state equation. The two parameter Mittag–Leffler function is defined as follows:

**Definition 3** (The two parameter Mittag–Leffler function)**.**

$$
E_{\alpha,\beta}(x) = \sum_{k=0}^\infty \frac{x^k}{\Gamma(k\alpha + \beta)}.
\tag{3}
$$

For $\beta = 1$ we obtain the one parameter Mittag–Leffler function:

**Definition 4** (The one parameter Mittag–Leffler function)**.**

$$
E_\alpha(x) = \sum_{k=0}^\infty \frac{x^k}{\Gamma(k\alpha + 1)}.
\tag{4}
$$

The fractional order, integro-differential operator (1) is described by different definitions, given by Grünwald and Letnikov (GL definition), Riemann and Liouville (RL definition) and Caputo (C definition). Relations between Caputo and Riemann-Liouville, between Riemann-Liouville and Grünwald-Letnikov operators are given, e.g., in [1,22]. Discrete versions of these operators are analysed with details in [23]. The C definition has a simple interpretation of an initial condition (it is analogical as in integer order case) and intuitive Laplace transform. Additionally, its value from a constant equals to zero, in contrast to, e.g., RL definition. That is why in the further consideration the C definition along time will be used. It is recalled beneath.

**Definition 5** (The Caputo definition of the FO operator).

$$
{}_0^C D_t^\alpha f(t) = \frac{1}{\Gamma(M - \alpha)} \int_0^\infty \frac{f^{(M)}(\tau)}{(t - \tau)^{\alpha + 1 - M}} d\tau.
\tag{5}
$$

In (5) $M$ is a limiter of the non integer order: $M - 1 \leq \alpha < M$. If $M = 1$ then, consequently, $0 \leq \alpha < 1$ is considered and the definition (5) takes the form:

$$
{}_0^C D_t^\alpha f(t) = \frac{1}{\Gamma(1 - \alpha)} \int_0^\infty \frac{\dot{f}(\tau)}{(t - \tau)^\alpha} d\tau.
\tag{6}
$$

Finally a fractional linear state equation using Caputo definition should be recalled. It is as follows:

$$
\begin{aligned}
{}_0^C D_t^\alpha x(t) &= Ax(t) + Bu(t) \\
y(t) &= Cx(t)
\end{aligned}
\tag{7}
$$

where $\alpha \in (0, 1)$ is the fractional order of the state equation, $x(t) \in \mathbb{R}^N$, $u(t) \in \mathbb{R}^L$, $y(t) \in \mathbb{R}^P$ are the state, control and output vectors respectively, $A, B, C$ are the state, control, and output matrices, respectively.

### 3. The Experimental System and Its FO Model

Figure 1 shows the simplified scheme of the considered heat system. It has the form of thin rectangular metallic surface (PCB plate) heated by flat heater, denoted by $H$. Coordinates of heater are denoted by $x_{h1}$, $x_{h2}$, $y_{h1}$ and $y_{h2}$, respectively. The temperature is read using thermal camera; the area of measurement is configurable. The area of measurement is denoted by $S$ and its coordinates are equal $x_{s1}$, $x_{s2}$, $y_{s1}$ and $y_{s2}$, respectively. More details about the construction of this laboratory system are given in the section "Experimental Results".

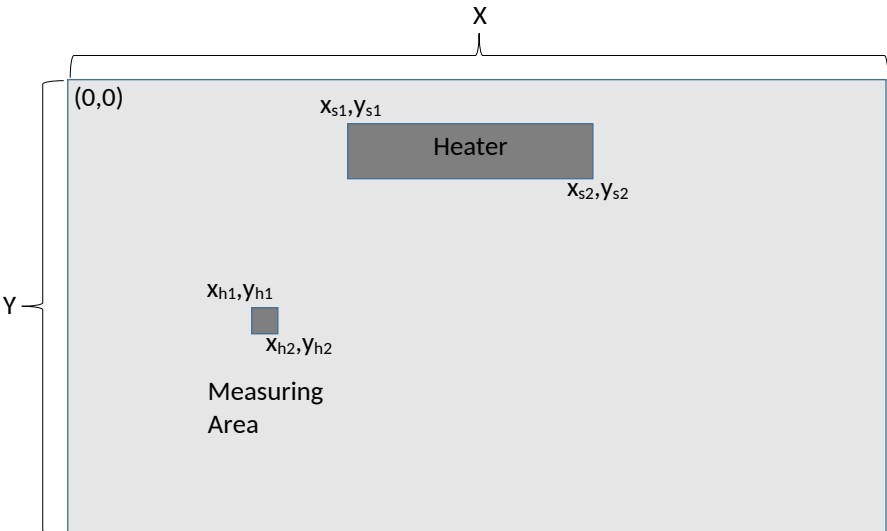

**Figure 1.** The simplified scheme of the experimental system. origin of the coordinate system is located in the left upper corner.

The fundamental mathematical model describing the heat transfer in the surface is the Partial Differential Equation (PDE) of the parabolic type. All the side surfaces of plate are much smaller than its frontal surface. This allows to assume the homogeneous Neumann boundary conditions at all edges of the plate as well as the heat exchange on the surface needs to be considered too. It is expressed by coefficient $R_a$. The control and observation

are distributed due to the size of heater and size of temperature field read by camera. The heat conduction along both directions $x$ and $y$ is the same and described by coefficient $a_w$. Two dimensional, IO heat transfer equation has been considered in many papers (e.g., [24–26]). Its fractional version proposed by authors is as beneath:

$$
\begin{cases}
{}_0^C D_t^\alpha Q(x,y,t) = a_w \left( \dfrac{\partial^2 Q(x,y,t)}{\partial x^2} + \dfrac{\partial^2 Q(x,y,t)}{\partial y^2} \right) - \\
- R_a Q(x,y,t) + b(x,y)u(t), \\
\dfrac{\partial Q(0,y,t)}{\partial x} = 0, t \geq 0 \\
\dfrac{\partial Q(X,y,t)}{\partial x} = 0, t \geq 0 \\
\dfrac{\partial Q(x,0,t)}{\partial y} = 0, t \geq 0 \\
\dfrac{\partial Q(x,Y,t)}{\partial y} = 0, t \geq 0 \\
Q(x,y,0) = Q_0, 0 \leq x \leq X, 0 \leq y \leq Y \\
y(t) = k_0 \int\limits_0^X \int\limits_0^Y Q(x,y,t)c(x,y)dxdy.
\end{cases}
\tag{8}
$$

In (8), $\alpha$ is non integer order of the system, $a_w > 0$, $R_a \geq 0$ are coefficients of heat conduction and heat exchange, $k_0$ is a steady-state gain of the model, $b(x,y)$ and $c(x,y)$ are heater and sensor functions. Denote the area of heater by $H$ and the area of measurement by $S$. Then the heater and sensor functions take the following form:

$$
b(x,y) = \begin{cases} 1, x, y \in H \\ 0, x, y \notin H \end{cases}.
\tag{9}
$$

$$
c(x,y) = \begin{cases} 1, x, y \in S \\ 0, x, y \notin S \end{cases}.
\tag{10}
$$

Let $\Omega \subset \mathbb{R}^N$ it be an appropriate restricted area. The Laplace operator $\Delta = \dfrac{\partial^2(..)}{\partial x^2} + \dfrac{\partial^2(..)}{\partial y^2}$ in $L^2(\Omega)$ with Dirichlet or Neumann boundary conditions is a discrete operator. The discrete operator has only a point spectrum (see, e.g., [27], pp. 204, 460). Without going into detail, it is generally known from the spectral theorem for compact and self-adjoint operators that all eigenvalues $\lambda_{m,n}$ of the Laplace operator $\Delta$ in $L^2(\Omega)$ (with Dirichlet or Neumann boundary conditions) are non-negative, with finite multiplicities and $\lambda_n \to \infty$ for $n \to \infty$. Additionally, there is in $L^2(\Omega)$ an orthonormal basis (complete system) composed of eigenfunctions of the appropriate operator $\Delta$. In special cases of the area $\Omega \subset \mathbb{R}^N$ (e.g., for a rectangle on a plane) analytical formulae for eigenvalues and eigenfunctions of the appropriate Laplace operator $\Delta$ can be given (see, e.g., [28], pp. 21, 26 for the Dirichlet problem or [29], pp. 133, 138, 301, 305).

However in the considered case the construction of the experimental system requires to assume the homogenous Neumann boundary conditions. Then eigenfunctions and eigenvalues are as follows:

$$
w_{m,n}(x,y) = \begin{cases}
1, \; m, n = 0, \\
\dfrac{2Y}{\pi n} \cos \dfrac{n\pi y}{Y}, \; m = 0, n = 1, 2, \ldots \\
\dfrac{2X}{\pi m} \cos \dfrac{m\pi x}{X}, \; n = 0, m = 1, 2, \ldots \\
\dfrac{2}{\pi} \dfrac{1}{\sqrt{\dfrac{m^2}{X^2} + \dfrac{n^2}{Y^2}}} \cos \dfrac{m\pi x}{X} \cos \dfrac{n\pi y}{Y}, \; m, n = 1, 2, \ldots
\end{cases}
\tag{11}
$$

$$
\lambda_{m,n} = -a_w \left[ \dfrac{m^2}{X^2} + \dfrac{n^2}{Y^2} \right] \pi^2 - R_a, \; m, n = 0, 1, 2, ..
\tag{12}
$$

Consequently, the 2D heat Equation (8) can be expressed as an infinite dimensional state equation:

$$\begin{cases} {}_0^C D_t^\alpha Q(t) = AQ(t) + Bu(t) \\ y(t) = CQ(t) \end{cases}. \tag{13}$$

where:

$$\begin{aligned} AQ &= a_w \left( \frac{\partial^2 Q(x,y)}{\partial x^2} + \frac{\partial^2 Q(x,y)}{\partial y^2} \right) - R_a Q(x,y), \\ D(A) &= \{ Q \in H^2(0,1) : Q'(0) = 0, Q'(X) = 0, Q'(Y) = 0 \}, \\ a_w, R_a &> 0, \\ CQ(t) &= < c, Q(t) >, Bu(t) = bu(t). \end{aligned} \tag{14}$$

The state vector $Q(t)$ is defined as beneath:

$$Q(t) = [q_{0,0}, q_{0,1}, q_{0,2} \ldots, q_{1,1}, q_{1,2}, \ldots]^T. \tag{15}$$

The state operator $A$ takes the following form:

$$A = diag\{\lambda_{0,0}, \lambda_{0,1}, \lambda_{0,2}, \ldots, \lambda_{1,1}, \lambda_{1,2}, \ldots, \lambda_{2,1}, \lambda_{2,2} \ldots, \lambda_{m,n}, \ldots\}. \tag{16}$$

The control operator $B$ is as beneath:

$$B = [b_{0,0}, b_{0,1}, \ldots, b_{1,0}, b_{1,1}, \ldots]^T. \tag{17}$$

where:

$$b_{m,n} = < H, w_{m,n} > = \int\limits_0^X \int\limits_0^Y b(x,y) w_{m,n}(x,y) dx dy. \tag{18}$$

Taking into account (11) we obtain:

$$b_{m,n} = \begin{cases} (x_{h2} - x_{h1})(y_{h2} - y_{h1}) \; m, n = 0, \\ \frac{1}{h_{yn}}(x_{h2} - x_{h1})\left(\sin(h_{yn}y_{h2}) - \sin(h_{yn}y_{h1})\right) \\ m = 0, n = 1, 2, 3, \ldots, \\ \frac{1}{h_{xm}}(y_{h2} - y_{h1})\left(\sin(h_{xm}x_{h2}) - \sin(h_{xm}x_{h1})\right) \\ n = 0, m = 1, 2, 3, \ldots, \\ \frac{1}{h_{xm}h_{yn}}\left(\sin(h_{yn}y_{h2}) - \sin(h_{yn}y_{h1})\right) \cdot \\ \cdot\left(\sin(h_{xm}x_{h2}) - \sin(h_{xm}x_{h1})\right) \\ m, n = 1, 2, 3, \ldots \end{cases} \tag{19}$$

where:

$$\begin{aligned} h_{xm} &= \frac{m\pi}{X}, \\ h_{yn} &= \frac{n\pi}{Y}. \end{aligned} \tag{20}$$

The output operator $C$ is defined analogically:

$$C = [c_{0,0}, c_{0,1}, \ldots, c_{1,0}, c_{1,1}, \ldots]. \tag{21}$$

where:

$$c_{m,n} = < S, w_{m,n} > = \int\limits_0^X \int\limits_0^Y c(x,y) w_{m,n}(x,y) dx dy. \tag{22}$$

Each element $c_{m,n}$ is expressed analogically, as (19):

$$c_{m,n} = \begin{cases} (x_{s2} - x_{s1})(y_{s2} - y_{s1}) \ m,n = 0, \\ \frac{1}{h_{yn}}(x_{s2} - x_{s1})(\sin(h_{yn}y_{s2}) - \sin(h_{yn}y_{s1})) \\ m = 0, n = 1,2,3,\ldots, \\ \frac{1}{h_{xm}}(y_{s2} - y_{s1})(\sin(h_{xm}x_{s2}) - \sin(h_{xm}x_{s1})) \\ n = 0, m = 1,2,3,\ldots, \\ \frac{1}{h_{xm}h_{yn}}(\sin(h_{yn}y_{s2}) - \sin(h_{yn}y_{s1})) \cdot \\ \cdot (\sin(h_{xm}x_{s2}) - \sin(h_{xm}x_{s1})) \\ m,n = 1,2,3,\ldots \end{cases} \tag{23}$$

In (23) $h_{xm,yn}$ are expressed by (20).

*3.1. The Decomposition of the Model*

The form of state operator $A$ expressed by (16) allows to decompose the system (13)–(23) to infinite number of independent scalar subsystems, associated to particular eigenvalues. This can be expressed as follows:

$$D^\alpha Q(t) = AQ(t) + Bu(t)$$
$$Aw_{m,n} = \lambda_{m,n}w_{m,n}$$
$$D^\alpha Q(t) = a_w \left(\frac{\partial^2 Q}{\partial x^2} + \frac{\partial^2 Q}{\partial y^2}\right) - R_a Q + Bu$$
$$D^\alpha \sum_{m=0}^\infty \sum_{n=0}^\infty q_{m,n} = a_w \left(\sum_{m=0}^\infty \frac{\partial^2 q_{m,n}}{\partial x^2} + \sum_{n=0}^\infty \frac{\partial^2 q_{m,n}}{\partial y^2}\right) - \tag{24}$$
$$- R_a \sum_{m=0}^\infty \sum_{n=0}^\infty q_{m,n} + Bu$$
$$D^\alpha q_{m,n} = a_w \left(\frac{\partial^2 q_{m,n}}{\partial x^2} + \frac{\partial^2 q_{m,n}}{\partial y^2}\right) - R_a q_{m,n} + b_{m,n}u.$$

The form of Equation (24) implies the decomposition of the system (7) into systems related to single eigenvalues $\lambda_{m,n}$, $m,n = 0,1,2,\ldots$ This decomoposition allows to easily compute the step and impulse responses of the system as a sum of responses of particular modes. This is presented below.

**Remark 1.** *Consider the system described by (13)–(22). Its step response takes the following form:*

$$y_\infty(t) = \sum_{m=1}^\infty \sum_{n=1}^\infty y_{m,n}(t). \tag{25}$$

*where $m,n$-th mode of response is as follows:*

$$y_{m,n}(t) = \frac{E_\alpha(\lambda_{m,n}t^\alpha) - 1(t)}{\lambda_{m,n}} b_{m,n}c_{m,n}. \tag{26}$$

*In (26) $E_\alpha(..)$ is the one parameter Mittag–Leffler function (4), $\lambda_{m,n}$, $b_{m,n}$ and $c_{m,n}$ are expressed by (12), (18) and (22) respectively.*

**Proof.** The $m,n$-th mode of the model is expressed as follows:

$$_0^C D_t^\alpha q_{m,n}(t) = \lambda_{m,n}q_{m,n}(t) + b_{m,n}1(t). \tag{27}$$

The Laplace transform from (27) with homogenous initial condition is as beneath:

$$s_t^\alpha q_{m,n}(s) = \lambda_{m,n} q_{m,n}(s) + \frac{b_{m,n}}{s} \iff$$
$$\iff q_{m,n}(s) = \frac{b_{m,n}}{s(s^\alpha - \lambda_{m,n})}$$

(28)

The inverse Laplace transform from (28) (see e.g., [2], p. 11; [30], p. 253) yields:

$$q_{m,n}(t) = \frac{E_\alpha(\lambda_{m,n} t^\alpha) - 1(t)}{\lambda_{m,n}} b_{m,n}.$$

(29)

Multiplying (29) by $c_{m,n}$ gives directly (26) and the proof is completed. □

The steady-state response to the Heaviside function $1(t)$ is as follows:

**Remark 2.** *Consider the system described by (13)–(22). Its steady-state response to the Heaviside function $1(t)$ takes the following form:*

$$y^{ss} = - \sum_{m,n=0}^{\infty} \frac{b_{m,n} c_{m,n}}{\lambda_{m,n}}.$$

(30)

**Proof.** The Laplace transform from $m, n$-th mode of step response is given by (28). Its steady-state value is as beneath:

$$y_{m,n}^{ss} = \lim_{t \to \infty} y_{m,n}(t) = \lim_{s \to 0} s \frac{b_{m,n} c_{m,n}}{s(s^\alpha - \lambda_{m,n})} = - \frac{b_{m,n} c_{m,n}}{\lambda_{m,n}}.$$

(31)

The sum of elements $y_{m,n}^{ss}$ gives directly (30). □

The impulse response of the proposed model is described by the following remark:

**Remark 3.** *Consider the system described by (13)–(22). Its impulse response is as beneath:*

$$g_\infty(t) = \sum_{m=1}^{\infty} \sum_{n=1}^{\infty} g_{m,n}(t).$$

(32)

*where:*

$$g_{m,n}(t) = t^{\alpha-1} E_{\alpha,\alpha}(\lambda_{m,n} t^\alpha) b_{m,n} c_{m,n}.$$

(33)

*In (33) $E_{\alpha,\alpha}(..)$ denote two - parameter Mittag–Leffler function (3), $\lambda_{m,n}$, $b_{m,n}$ and $c_{m,n}$ are expressed by (12), (18) and (22) respectively.*

The above remark can be proved with the use of Laplace transform, analogically as (25).

The crucial difference to one-dimensional heat system presented in the paper [31] is that the eigenvalues can be multiple.

The existence of multiple eigenvalues is determined by the dimensions $X$ and $Y$ of the plate and it can be described by the following remark:

**Remark 4** (The existence of multiple eigenvalues in the model). *Consider the model of two dimensional heat transfer described by state equation with state operator (16). Assume that the eigenvalues of this state operator are expressed by (12) and the size of plate is equal $X \times Y$. Then two different eigenvalues $\lambda_{m_1,n_1}$ and $\lambda_{m_2,n_2}$ (where $m_1 \neq m_2$, $n_1 \neq n_2$) are equal iff:*

$$Y = X \sqrt{\frac{n_1{}^2 - n_2{}^2}{m_1{}^2 - m_2{}^2}},$$
$$m_1, m_2 = 0, 1, 2, \ldots, n_1, n_2 = 0, 1, 2, \ldots, m_1 \neq m_2, n_1 \neq n_2.$$

(34)

**Proof.** Taking into consideration (12) we obtain:

$$\lambda_{m_1,n_1} = -a_w \left[ \frac{m_1^2}{X^2} + \frac{n_1^2}{Y^2} \right] \pi^2 - R_a,$$

$$\lambda_{m_2,n_2} = -a_w \left[ \frac{m_2^2}{X^2} + \frac{n_2^2}{Y^2} \right] \pi^2 - R_a.$$

Comparing: $\lambda_{m_1,n_1} = \lambda_{m_2,n_2}$ and making some elementary transformations we obtain directly (34). □

Analogically, as in one dimensional case discussed, u.a., in [31] the model (13)–(33) is infinite dimensional. Its practical implementation requires to use finite dimensional approximation obtained via truncation of further modes of step or impulse responses. Then the infinite dimensional operators *A*, *B* and *C* can be interpreted as matrices. Consequently, step and impulse responses are the following finite sums:

$$y_{MN}(t) = \sum_{m=0}^{M} \sum_{n=0}^{N} y_{m,n}(t). \tag{35}$$

$$g_{MN}(t) = \sum_{m=0}^{M} \sum_{n=0}^{N} g_{m,n}(t). \tag{36}$$

In (35) and (36) *M* and *N* denote the size of finite dimensional approximation. It can be estimated numerically or analytically.

### 3.2. The Stability

The spectrum of the proposed model is defined as the set of all eigenvalues of the system:

$$\sigma = \{\lambda_{0,0}, \lambda_{0,1}, \ldots, \lambda_{1,0}, \ldots, \lambda_{m,n}, \ldots\}. \tag{37}$$

where $\lambda_{m,n}$ are defined by (12). The spectrum contains negative, single or multiple, pure real eigenvalues. This implies that the proposed FO model is asymptotically stable for each value of plant parameters $\alpha$, $a_w$ and $R_a$.

Another interesting problem is the Mittag–Leffler stability of the model. It is determined by location of eigenvalues in the left semiplane. In the one dimensional case all eigenvalues in the spectrum are ordered in the decreasing order. The most poorly damped eigenvalue is always most close to imaginary axis, next eigenvalues follow it in the decreasing order.

In the two dimensional case, we deal with the situation is rather more complicated due to the value of $\lambda_{m,n}$ is determined by two indices *m* and *n* as well as dimensions of the plate. The most poorly damped eigenvalue is still the eigenvalue $\lambda_{0,0}$, but next eigenvalues are not ordered in the decreasing order. Determing the damping rate requires to order all others eigenvalues.

### 3.3. The Convergence

The convergence of the proposed model will be tested using approach close to presented in papers [32,33]. This can be performed by estimating the orders *M* and *N* assuring a predefined value of Rate Of Convergence (ROC). In the considered case the ROC is defined as the increment of steady-state response $y_{m,n}^{ss}$ as a function of orders *M* and *N*. This increment is equal to the absolute value of *M*, *N*-th mode of the steady-state response (31):

$$ROC(M, N) = |y_{MN}^{ss}|. \tag{38}$$

The *ROC* as a function of orders *M* and *N* is expressed by the following remark:

**Remark 5.** *Consider the system* (13)–(22). *The Rate of Convergence (ROC) as a function of both orders M and N is expressed as beneath:*

$$ROC(M, N) = \left| \frac{H_R P_h P_s}{\lambda_{M,N}} \right|. \tag{39}$$

*where $\lambda_{M,N}$ is expressed by* (12), *the coefficients $H_R$, $P_h$ and $P_s$ are as follows:*

$$
\begin{aligned}
H_R &= \left( \frac{XY}{MN\pi^2} \right)^2, \\
P_h &= \big(\sin(h_{yn}y_{h2}) - \sin(h_{yn}y_{h1})\big)\big(\sin(h_{xm}x_{h2}) - \sin(h_{xm}x_{h1})\big), \\
P_s &= \big(\sin(h_{yn}y_{s2}) - \sin(h_{yn}y_{s1})\big)\big(\sin(h_{xm}x_{s2}) - \sin(h_{xm}x_{s1})\big).
\end{aligned}
\tag{40}
$$

*In* (40) *$h_{xm}$ and $h_{yn}$ are described by* (20).

The form of $ROC(M, N)$, expressed by (39), is rather complicated. To simplify it assume the maximum values of $P_h$ and $P_s$ equal 4. This allows to give the upper estimation of (39) in the following form:

$$ROC(M, N) \leq \left| \frac{16H_R}{\lambda_{M,N}} \right|. \tag{41}$$

The values of orders $M$ and $N$ assuring the keeping a predefined value $\Delta$ of $ROC$ can be computed via numerical solution of the following inequality:

$$ROC(M, N) \leq \Delta. \tag{42}$$

## 4. Experimental Validation of Results

Experiments were conducted using laboratory system shown in the Figure 2. The dimensions of the PCB plate in pixels are following: $X = 380$, $Y = 290$. The PCB is heated by the heater $170 \times 20$ pixels with maximum power 10 W located in points: $x_{h1} = 100$, $y_{h1} = 40$ The temperature of the plate is measured using thermal camera OPTRIS PI 450, connected to computer via USB. The measured range of temperature is 0–250 °C, the frequency of sampling is 80 Hz. Data from camera are collected using dedicated software Optris PI Connect. The signal powering the heater is given from computer using NI LabView, NI MyRIO and amplifier. The maximum current from amplifier equal 400 [mA] at a voltage of 12 V gives a power of 4.8 W. The tested PCB plate not isolated from the environment. This implies that measurements strongly depend on ambient temperature. The presented experiment was conducted during the heat of summer.

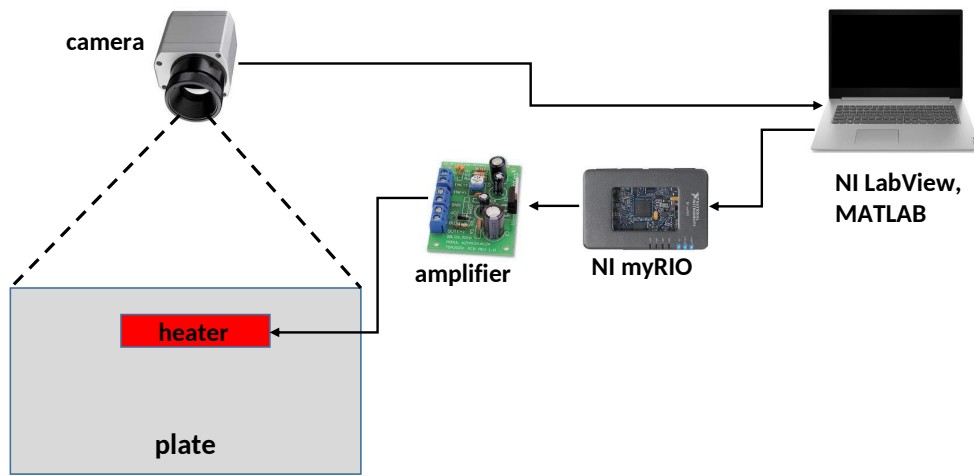

**Figure 2.** The experimental system.

Estimation the quality of the models only by comparing of diagrams is not enough accurate. The use of cost function is necessary, because only it allows to precisely estimate the accuracy of a model. Additionally such a cost function is employed to parameters identification. In this paper the Mean Square Error (MSE) cost function is used. It describes the mean difference between step response of plant and model at the same time-spatial mesh:

$$MSE = \frac{1}{K} \sum_{k=1}^{K} (y(k) - y_e(k))^2. \tag{43}$$

In (43), $K$ is the number of all collected samples, $y(k)$ is the step response of the model, computed using (35), $y_e(k)$ is the experimental response measured in the same place and at the same time instants $k$ with the use of thermal camera. The sample time during a step response measurement was equal 1 [s]. In each case the mean temperature of the whole area is measured.

During experiments, the step response of the system was investigated. The "zero" level denotes the heater switched off, the "one" level is the full power of the heater. The temperature fields for both states are shown in the Figure 3. This figure shows also the points of measurement of the step response, marked as "Area 1–4". Areas 1–3 are located in different points of plate, area 4 covers the heater and it describes its mean temperature. Coordinates of all measuring areas are described by the Table 1. During calculations these coordinates $x_{..}$ and $y_{..}$ were given relative to $X$ and $Y$. For example, $x_{s1} = 75$ during calculating elements of $C$ matrix with respect to (23) was equal: $x_{s1} = \frac{75}{380} = 0.1974$.

**Table 1.** Coordinates of measuring areas (in pixels).

| Area | $x_{s1}$ | $y_{s1}$ | $x_{s2}$ | $y_{s2}$ |
|:---:|:---:|:---:|:---:|:---:|
| 1 | 50 | 75 | 52 | 77 |
| 2 | 200 | 100 | 202 | 102 |
| 3 | 300 | 200 | 302 | 202 |
| 4 | 120 | 40 | 250 | 60 |

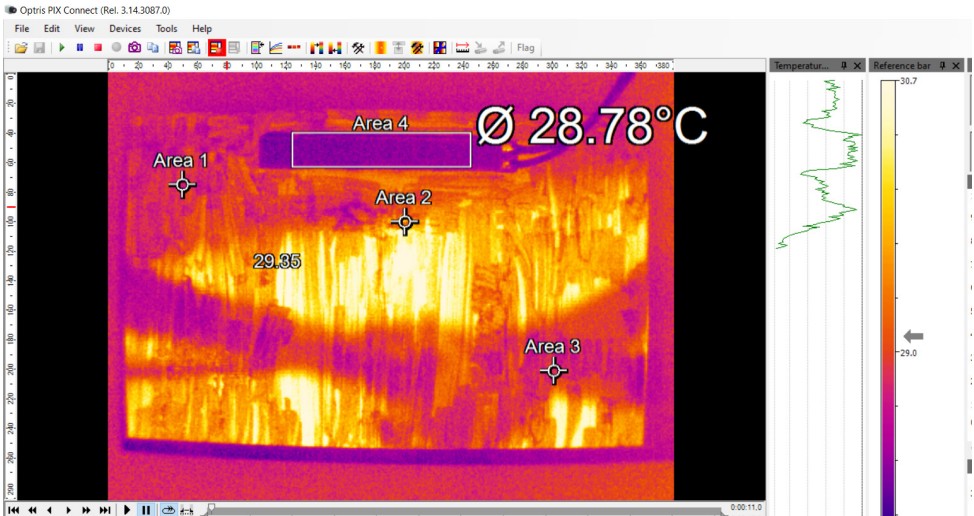

**Figure 3.** *Cont.*

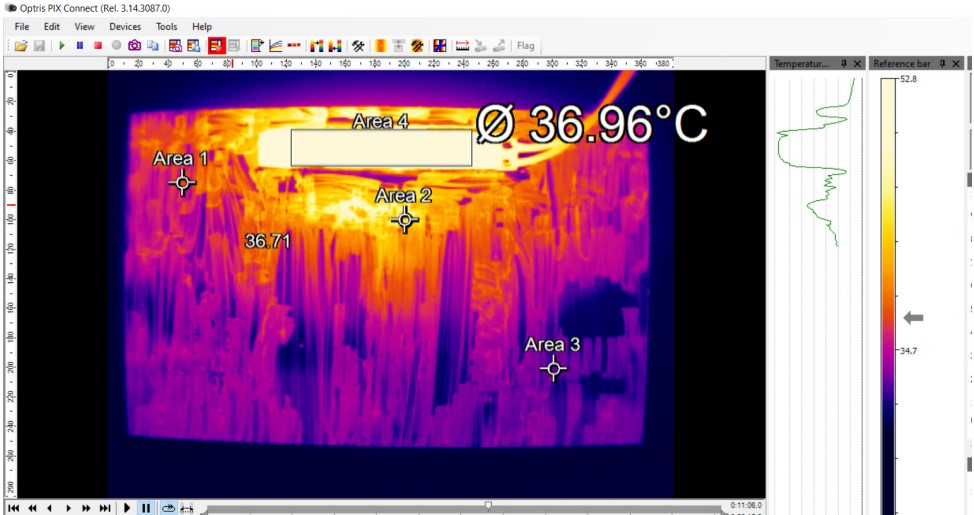

**Figure 3.** The steady-state temperature fields for non heated (**top**) and heated plate (**bottom**). The temperature strongly depends on ambient temperature. The colour scale in each case is different.

The step responses in the selected areas 1, 2, 3 and 4 are shown in the Figure 4.

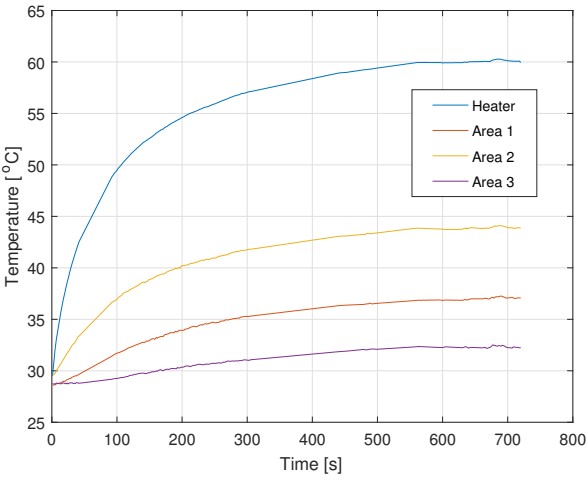

**Figure 4.** The step responses of temperature in all tested fields.

The parameters of the fractional model were identified via minimization the cost function (43) as a function of parameters $\alpha$, $a_w$ and $R_a$. Optimization was conducted with the use of MATLAB function *fminsearch*, the step response was calculated using Formula (35). Results of identification are given in the Table 2 and illustrated by Figures 5–8. The dimensions of model for all experiments were equal: $M = N = 8$ and this gives the summarized size of the applied model equal 64.

**Table 2.** Identified parameters of the fractional model.

| Area | $\alpha$ | $a_w$ | $R_a$ | Cost Function MSE (43) |
|------|----------|-------|-------|------------------------|
| 1 | 1.0794 | 0.0032 | 0.0032 | 0.0110 |
| 2 | 0.9356 | 0.0008 | 0.0090 | 0.0217 |
| 3 | 1.4877 | 0.0035 | 0.0003 | 0.0059 |
| 4 | 0.8156 | 0.0078 | 0.0235 | 0.0627 |

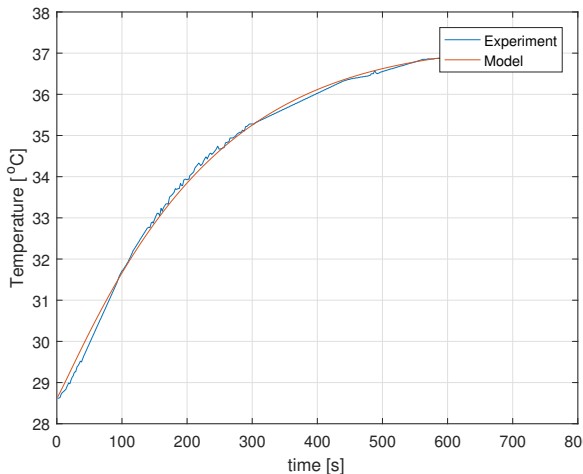

**Figure 5.** The step responses of the FO model and real plant for area 1.

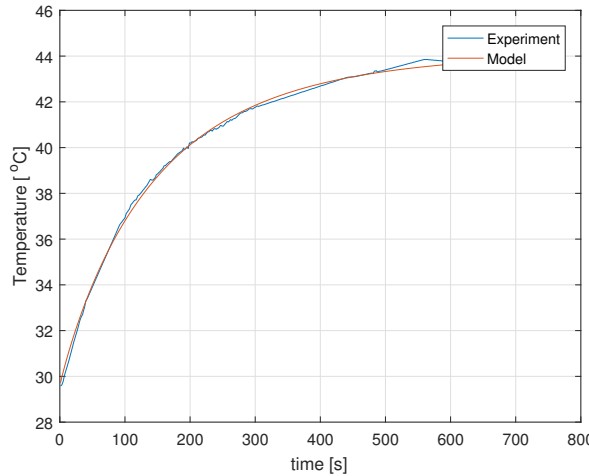

**Figure 6.** The step responses of the FO model and real plant for area 2.

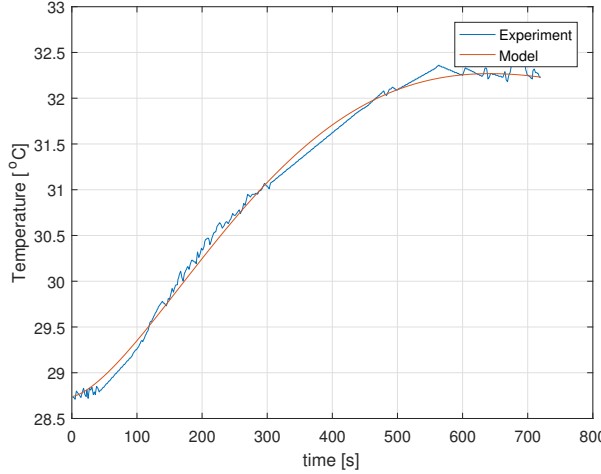

**Figure 7.** The step responses of the FO model and real plant for area 3.

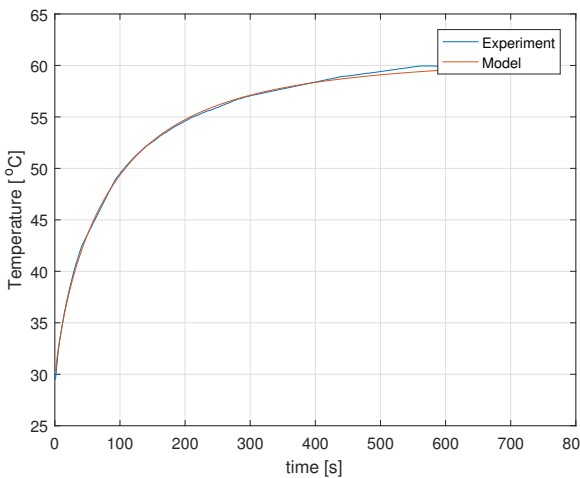

**Figure 8.** The step responses of the FO model and real plant for area 4.

Next, the accuracy of the proposed FO model was compared to accuracy of analogical Integer Order (IO) model, obtained for $\alpha = 1.0$. The step response of IO model was computed using MATLAB function *step*. Results are given in the Table 3 and illustrated by Figures 9–12.

**Table 3.** Identified parameters of the integer order model.

| Area | $a_w$ | $R_a$ | **Cost Function MSE** (43) |
|:---:|:---:|:---:|:---:|
| 1 | 0.0032 | 0.0045 | 0.0233 |
| 2 | 0.0033 | 0.0066 | 0.0497 |
| 3 | 0.0035 | 0.0028 | 0.0664 |
| 4 | 0.0038 | 0.0098 | 1.1448 |

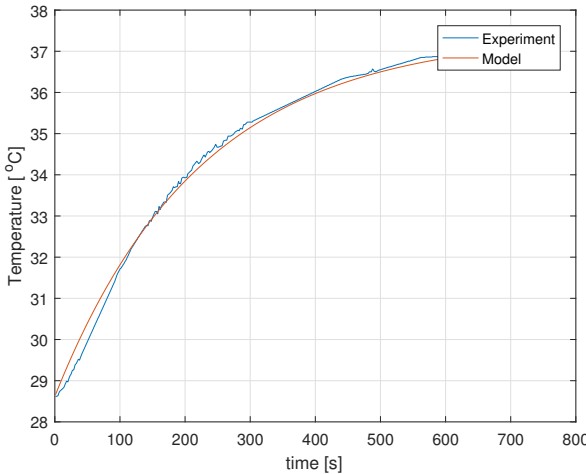

**Figure 9.** The step responses of the IO model and real plant for area 1.

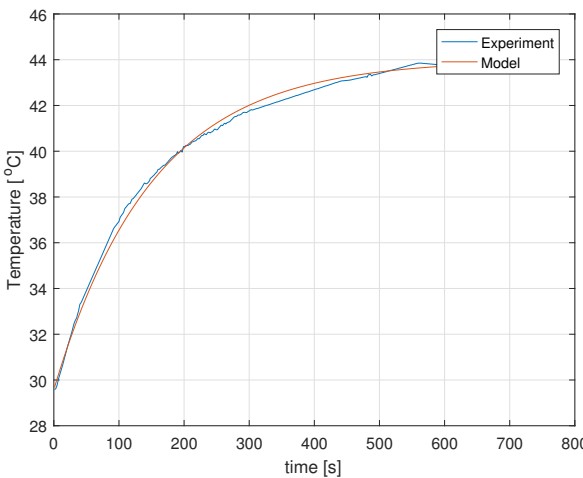

**Figure 10.** The step responses of the IO model and real plant for area 2.

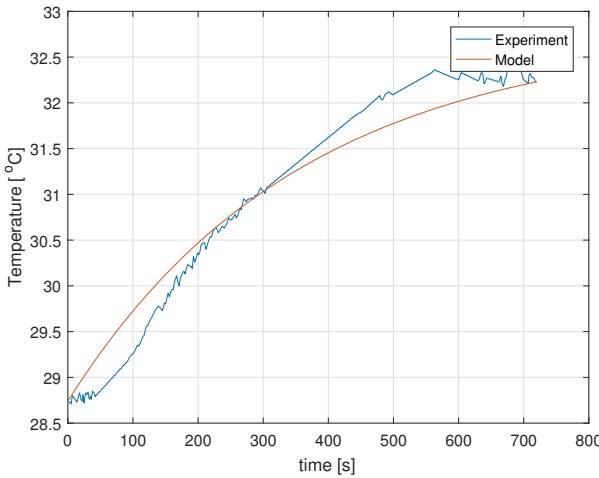

**Figure 11.** The step responses of the IO model and real plant for area 3.

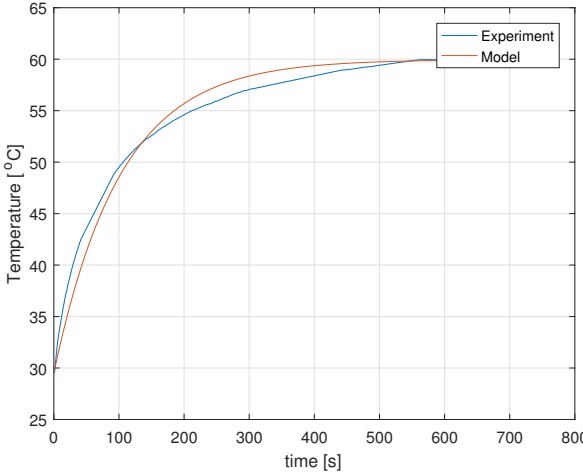

**Figure 12.** The step responses of the IO model and real plant for area 4.

## 5. Discussion of Results and Final Conclusions

At the beginning limitations of the proposed model need to be discussed.

Firstly, the proposed model uses fractional operator only to express derivative along the time and the derivative along the space coordinates is being still integer order. Results

for one dimensional heat transfer given in [34] point that the most accurate model uses fractional derivatives along time and space coordinates. On the other hand, the analysis of the proposed model, using only FO operator along time is simplier.

Next, the surface which the temperature is measured, is non homogenous in the sense of values of parameters $\alpha$, $a_w$ and $R_a$. This was observed during identification, when we obtained different values of parameters for different points of measurement. This is visible well in area 3. In addition, this point is the most distant from heater and the impact of disturbances is more significant than in other tested places. A more precise model should employ parameters dependent on spatial coordinates $x$ and $y$. Such a a model is planned to construct during further investigations.

An another limitation are dimensions $M$ and $N$ of the model. They can not be too high due to the summarized size of the model equal $MN$. Increasing of these dimensions causes increasing numerical complexity of the model. Fortunately, the proposed model assures relatively good accuracy for $M$, $N < 10$. The numerical complexity is also worth researching during future work.

Next, the shape of analyzed surface is relatively simple (rectangular). In reality, a temperature of more complicated elements needs to be modeled. This is not a trivial issue due to it requires to consider much more complicated form of border conditions.

Furthermore, some properties, e.g., positivity of this model need o be more deeply analyzed.For one dimensional case the positivity was analyzed with details in the paper [35] and results presented here need to be generalized to the two-dimensional plant considered here. This is planned to do during future work.

The main final conclusion from this paper is that the proposed FO model is more accurate in the sense of the considered MSE cost function than the analogical IO model. This is visible by comparing Table 2 to Table 3.

**Author Contributions:** Conceptualization, K.O.; methodology W.M. and K.O.; formal analysis W.M. and K.O.; software M.R. and K.O.; validation M.R.; experiments M.R.; supervision K.O. All authors have read and agreed to the published version of the manuscript.

**Funding:** This research was funded by AGH grant number 16.16.120.773.

**Conflicts of Interest:** The authors declare no conflict of interest.

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
