# Peer review of "Fractional Order Model of the Two Dimensional Heat Transfer Process"

_energies, doi:10.3390/en14196371_

Round 1

Reviewer 1 Report

The authors have explored a fractional-order model of a heat transfer in the two-dimensional plate. The spectrum decomposition and stability of the model are analyzed. An experiment is realized to verify the model.

The subject is interesting; however, I have some remarks that should be considered in the revised version:

The application of fractional calculus need to be discussed more at least in the introduction ( see for example Results in Physics 19, 103453; Fractals, 28 (2020) 2040035)

The authors have used the Caputo definition, it will be better for the reader to clarify why is used this type of fractional calculus.

The authors should include more details on the experiment in particular the numerical values used.

The authors should discuss the limitation of their model.

They also need to add more details in the captions of the figures

Author Response

Tnak You for comments and suggestions. My answers are below.

The authors have explored a fractional-order model of a heat transfer in the two-dimensional plate. The spectrum decomposition and stability of the model are analyzed. An experiment is realized to verify the model.

The subject is interesting; however, I have some remarks that should be considered in the revised version:

The application of fractional calculus need to be discussed more at least in the introduction ( see for example Results in Physics 19, 103453; Fractals, 28 (2020) 2040035)

RE:

Clear, positions were added and cited.

The authors have used the Caputo definition, it will be better for the reader to clarify why is used this type of fractional calculus.

RE:

The Caputo definition has a simple interpretation of an initial condition (it is analogical as in integer order case) and intuitive Laplace transform. Additionally its value from a constant equals to zero, in contrast  to e.g. RL definition.

This explaination  was added before C definition (eq 5).

The authors should include more details on the experiment in particular the numerical values used.

RE:

Some information were added.

The authors should discuss the limitation of their model.

RE:

Of course. The discussion of limitations was added in final conclusions.

They also need to add more details in the captions of the figures

RE:

Some information was added in the captions, however crucial inofmration is given in the text.

Reviewer 2 Report

Review under the paper â„–1326864

Fractional order model of the two dimensional heat transfer process

Authors: Krzysztof Oprzedkiewicz, Wojciech Mitkowski, Maciej RosóÅ‚.

   The authors considered the problem of heat transfer in quasi-two-dimensional structures with numerical modeling of its solution, which they illustrated graphically. In this regard, I would like to note that in the works of other authors similar problems were solved and, in particular, the processes of heat transfer in quasi-one-dimensional structures and quasi-dimensional ones were described.

At the same time, it is worth paying attention to the fact that the use of the fractional differentiation operator in the form of the Riemann-Liouville integral, which is used by the authors of the peer-reviewed article, does not always allow us to come to an analytical form of recording the final answer.

In this regard, the authors need to comment on the answer they received, and explain its difference from the answer that could be obtained in the framework of the application of fractional differentiation according to Fourier, which was used by the famous mathematician Letnikov.

   The authors need to take this note into account, as well as significantly expand the rather meager bibliographic list, adding to it a number of articles on the subject of fractional differentiation in the form of the Fourier integral used by other authors who were not mentioned in their work.

In general, the article makes a good impression and could be of interest to the readership of Energies journal.  

   After taking into account these comments and serious revision after a secondary review, the article can be published in the journal Energies. 

Author Response

Thank You for comments. My answers are below.

The authors considered the problem of heat transfer in quasi-two-dimensional structures with numerical modeling of its solution, which they illustrated graphically. In this regard, I would like to note that in the works of other authors similar problems were solved and, in particular, the processes of heat transfer in quasi-one-dimensional structures and quasi-dimensional ones were described.

At the same time, it is worth paying attention to the fact that the use of the fractional differentiation operator in the form of the Riemann-Liouville integral, which is used by the authors of the peer-reviewed article, does not always allow us to come to an analytical form of recording the final answer.

In this regard, the authors need to comment on the answer they received, and explain its difference from the answer that could be obtained in the framework of the application of fractional differentiation according to Fourier, which was used by the famous mathematician Letnikov.

RE:

In our paper the Caputo operator is employed. This allows to avoid problems mentioned above.

   The authors need to take this note into account, as well as significantly expand the rather meager bibliographic list, adding to it a number of articles on the subject of fractional differentiation in the form of the Fourier integral used by other authors who were not mentioned in their work.

RE:

Suitable  positions were addded and cited.

In general, the article makes a good impression and could be of interest to the readership of Energies journal.  

RE:

Thank You!

Reviewer 3 Report

This paper presents a state space, fractional order model of a heat transfer in a two dimensional plate.

The  manuscript is well written and the maths seems correct.

The experimental results and the comparisons with the integer order model are interesting.

Analyzing a further 2D geometry could improve this paper strenght.

Author Response

Thank You for comments. My answers are below.

This paper presents a state space, fractional order model of a heat transfer in a two dimensional plate.

The  manuscript is well written and the maths seems correct.

The experimental results and the comparisons with the integer order model are interesting.

Analyzing a further 2D geometry could improve this paper strenght.

RE:

Thank You!

The analysis of another shapes of measured surface are planned. This was added in the final discussion.

Reviewer 4 Report

In this paper, the authors considered a new, state-space, fractional-order model of heat transfer in a two-dimensional plate. The proposed model is derived directly from a two-dimensional heat transfer equation by employing the Caputo operator to express the fractional-order differences over time. The spectrum decomposition and stability of the model are analyzed. The formulae of impulse and step responses of the model are proved. Theoretical results are verified using experimental data from the thermal camera. The comparison of the theoretical model with the experiment shows that the proposed fractional model is more accurate than the integer-order model (in the sense of MSE cost function).

Observations:

1). Eq. (8). In the last line must be “dxdy”.

2). Verify Eqs. (9) and (10). What are the significations of “H” and “S”?

3). Page 7. Eq. (26) is not correct. The demonstration contains mistakes.

For the inverse Laplace transform of (28) you must use the formula

Inverse Laplace of{1 / s(s^alpha – b)}=(1/b)[E_alpha(bt^alpha) – 1]. Therefore, the argument of the Mittag-Leffler function must contain “t” at the power “alpha”.

The same mistake is in eq. (33).

For the Laplace transforms of the Mittag-Leffler functions, see for example

https://link.springer.com/content/pdf/bbm%3A978-1-4471-2233-3%2F1

4). Because you use the results regarding the inverse Laplace transform in your analyzes, the above mistakes affect subsequent results.

Author Response

Thank You for review and comments. My answers are below.

Comments and Suggestions for Authors

In this paper, the authors considered a new, state-space, fractional-order model of heat transfer in a two-dimensional plate. The proposed model is derived directly from a two-dimensional heat transfer equation by employing the Caputo operator to express the fractional-order differences over time. The spectrum decomposition and stability of the model are analyzed. The formulae of impulse and step responses of the model are proved. Theoretical results are verified using experimental data from the thermal camera. The comparison of the theoretical model with the experiment shows that the proposed fractional model is more accurate than the integer-order model (in the sense of MSE cost function).

Observations:

1). Eq. (8). In the last line must be “dxdy”.

RE:

Corrected

2). Verify Eqs. (9) and (10). What are the significations of “H” and “S”?

RE:

Clear. H denotes the area of heater and S denotes the area of measurement. Both elements are marked in the scheme in figure 1, explainations were added in the text.

3). Page 7. Eq. (26) is not correct. The demonstration contains mistakes.

For the inverse Laplace transform of (28) you must use the formula

Inverse Laplace of{1 / s(s^alpha – b)}=(1/b)[E_alpha(bt^alpha) – 1]. Therefore, the argument of the Mittag-Leffler function must contain “t” at the power “alpha”.

The same mistake is in eq. (33).

For the Laplace transforms of the Mittag-Leffler functions, see for example

https://link.springer.com/content/pdf/bbm%3A978-1-4471-2233-3%2F1

RE:

Thank for very important remark! Of course! Corrections were made in equations (26) and (33) as well as in (29). The linked book was added to bibliography and cited.  

4). Because you use the results regarding the inverse Laplace transform in your analyzes, the above mistakes affect subsequent results.

RE:

Of course, all identifications were done second time with the use of corrected equations. Results are given in the revised version. The correction allowed to significantly improve the quality of the model.

Reviewer 5 Report

As it can be understood from the title, the study is about Fractional order model of the two dimensional heat transfer process. My research field is  a little bit far from this field, but following points took my attention, 

a) English should be improved , e.g.  in page 2: "in two dimensional plate. Such a process "

b) When the Figure and Table words referring to a figure or table should be started by capital letter. 

c) Validation parts should be explained in details, putting figures is not sufficient.

d) Why is the agreement between FO model and experiment for area 3 is not good compared to other areas. 

e) As I see, the results of FO and IO models are the same , so the only difference is  MSE cost function? Is it correct?

f) The conclusion part is not appropriate, it should be supported by data and more explanations are required.  We have a paper with 18 pages but conclusion is 3 line. I think it is not good. 

Author Response

Thank You for comments, my answers are below.

As it can be understood from the title, the study is about Fractional order model of the two dimensional heat transfer process. My research field is  a little bit far from this field, but following points took my attention, 

  1. a) English should be improved , e.g.  in page 2: "in two dimensional plate. Such a process "

RE:

English was tried to correct in the whole text.

  1. b) When the Figure and Table words referring to a figure or table should be started by capital letter. 

RE:

Corrected

  1. c) Validation parts should be explained in details, putting figures is not sufficient.

RE:

This was tried to do. Information about experiment was expanded.

  1. d) Why is the agreement between FO model and experiment for area 3 is not good compared to other areas. 

RE:

  1. e) As I see, the results of FO and IO models are the same , so the only difference is  MSE cost function? Is it correct?

RE:

The estimation of the quality of a model only by comparing of diagrams is not enough accurate. The use of cost function is necessary, because only the cost function allows to precisely estimate the accuracy of a model.

  1. f) The conclusion part is not appropriate, it should be supported by data and more explanations are required.  We have a paper with 18 pages but conclusion is 3 line. I think it is not good. 

 RE:

Clear, the final disussion and the conclusions were expanded.

Round 2

Reviewer 1 Report

The revised version is ok. I accept it.

Author Response

Thank You!

Reviewer 2 Report

   The authors did not take into account the observation regarding the comparison of their results with the results obtained by other authors on this topic.

Author Response

   The authors did not take into account the observation regarding the comparison of their results with the results obtained by other authors on this topic.

RE:

Thank You for suggestions. The exact form of the issue presented by us has not been previously published.

Similar problems were solved in some papers, citations were added. Changes are marked in blue in the revised text.

Reviewer 3 Report

The authors aswered to my suggestion. However, the analysis of a further 2D geometry was only planned.

Author Response

The authors aswered to my suggestion. However, the analysis of a further 2D geometry was only planned.

RE:

Thank You for remark. Analysis of the another shape of surface is not a trivial issue. Especially, an another form of border conditions needs to be proposed. This is mentioned in the final discussion.

Reviewer 4 Report

In the revised manuscript, the authors made the requested corrections.

Author Response

Thank You!

Reviewer 5 Report

I have two requests and then I can judge the revised paper better .

a) Is it possible to write the changes in the revised paper in other color. I could not find which parts of the paper have been revised. 

b) I did not receive any answers for the following question: 

  1. d) Why is the agreement between FO model and experiment for area 3 is not good compared to other areas. 

Thank you 

Author Response

Is it possible to write the changes in the revised paper in other color. I could not find which parts of the paper have been revised. 

RE:

OK, all the corrections with respect to Your comments were marked in green

I did not receive any answers for the following question: 

d) Why is the agreement between FO model and experiment for area 3 is not good compared to other areas

RE:

Thank You. I am sorry . The answer is that the area 3 is the most distant from heater ant the impact of disturbances is more significant than in other tested places. This is mentioned in the final discussion.

Round 3

Reviewer 2 Report

The authors discussed in more detail the results obtained by them and the quality of the article improved.

Reviewer 5 Report

I think that this paper can be accepted in the present form and no need for further revision.